

# Effects of functional correction training on movement patterns and physical fitness in male college students

Zhiyong Zhang[1],*, Lunxin Chen[1],*, Ziqing Qin[1], Jiaxin He[2], Chong Gao[2], Jian Sun[2], Jiancai Chen[2] and Duanying Li[2]

[1] Digitalized Performance Training Laboratory, Guangzhou Sport University, Guangzhou, Guangdong, China
[2] School of Physical Education, Guangzhou Sport University, Guangzhou, Guangdong, China
* These authors contributed equally to this work.

Corresponding authors
Jiancai Chen,
gtchenjiancai@163.com
Duanying Li,
liduany@gzsport.edu.cn

## ABSTRACT

The objective of this study is to investigate the effects of functional corrective training and static stretching on the quality of movement patterns and physical fitness in college students. The study was conducted with 30 male college students from a university in Guangzhou, China. The participants were randomly assigned to either the functional corrective training group (FCT, $n$ = 15, age = 20.93 ± 0.85, BMI = 22.07 ± 2.33) or the static stretching group (SS, $n$ = 13, age = 20.85 ± 0.86, BMI = 21.98 ± 1.80). Two participants from the SS group dropped out due to personal reasons, leaving 13 subjects in that group. Both groups underwent a 6-week training intervention, with sessions held twice a week. The FCT group participated in flexibility training, and/or static motor control training, and/or dynamic motor control training for 10–15 min. The SS group performed static stretching exercises targeting five specific muscles, with 30 s per side and two sets. The Functional Movement Screen (FMS), body composition, sit-and-reach, standing long jump, and pull-ups were assessed before and after the intervention. Differences in FMS outcomes were analyzed using two samples of the Mann-Whitney U test. Physical fitness outcomes were analyzed using a repeated measures analysis of variance (ANOVA) with a 2 (group) × 2 (time) design. After 6 weeks of intervention, the FCT group showed statistically significant improvements in the hurdle step ($Z$ = −2.449, $p$ = 0.014), inline lunge ($Z$ = −2.000, $p$ = 0.046), rotary stability ($Z$ = −2.309, $p$ = 0.021), and composite scores ($Z$ = −3.316, $p$ = 0.001). Comparisons between groups indicated that BMI (FCT, ES = 0.04; SS, ES = −0.11), 30-m sprint (FCT, ES = 0.12; SS, ES = 0.28), body fat percentage (BF%) (FCT, ES = −0.25; SS, ES = −0.07), and sit-and-reach (FCT, ES = 0.17; SS, ES = 0.06) were not statistically significant in both the pre- and post-tests. The effect sizes of all physical fitness indicators were greater in the FCT group than in the SS group. The FCT, consisting of two sessions per week for 6 weeks, has been proven to be effective in improving the quality of movement patterns by improved stability and advanced movements. However, the improvements in physical fitness did not reach statistical significance. FMS and FCT are generally affordable and accessible for college students. College students have the opportunity to employ the FMS tool to assess potential injury risks and address them, thereby reducing the risk of injuries.

# INTRODUCTION

According to the 2021 sampling review conducted by the Chinese Ministry of Education on student physical fitness, college students had the highest failure rate at 30.0% among students in various grades nationwide as of 2020 (*People's Daily, 2020*). This may be attributed to their lack of physical activities (*Pengpid et al., 2015*). Research indicates that 48.19% of college students in China participate in physical activity less than three times a week, and 58.7% spend less than 30 min per activity (*Song, 2023*). Several factors, including restricted access to facilities, academic stress, and a lack of professional exercise guidance, may contribute to this lack of physical activity (*Pan et al., 2022*). Sports injuries are also prevalent among Chinese college students, with a prevalence ranging from 24.0 to 31.0% (*Tang et al., 2020*; *Wu et al., 2019*). Furthermore, studies have shown that a significant proportion of college students demonstrate asymmetry in Functional Movement Screen (FMS) and achieve a composite score of ≤14, both of which indicate a high risk of injuries (*Bonazza et al., 2017*; *Mokha, Sprague & Gatens, 2016*). The prevalence of such scores varies, with percentages ranging from 53.9% among student-athletes to 57.3% among general college students. (*Engquist et al., 2015*). Additionally, according to *Triplett et al. (2021)*, approximately 57% of college students exhibit at least one asymmetry. These findings highlight the urgent need to prioritize enhancing physical fitness and addressing the FMS among college students. This will promote their overall well-being and reduce the risk of injuries.

The FMS, developed by *Cook et al. (2010)* is an assessment and training system that is based on the development of functional movement. The core concept of the FMS is "good movement quality before progressing to more movements" (*Cook, 2003*). The system consists of the FMS and functional corrective training (FCT) (*Cook, 2003*; *Cook et al., 2010*). Research indicates that injuries are more likely with incorrect movement patterns (*Cook et al., 2014b*). The FMS reliably identifies potential injury factors such as asymmetry, pain, and dysfunction (FMS sub-score < 1) (*Bonazza et al., 2017*; *Schneiders et al., 2011*; *Smith et al., 2013*). The FMS includes seven movement patterns tests: deep squat, hurdle step, inline lunge, shoulder mobility, active straight leg raise, trunk stability push-up, and rotary stability. Each test is scored on a scale of 0 to 3, with a total possible score of 21, assessing asymmetry, pain, and dysfunction (*Cook et al., 2014a*, *2014b*). It is recommended to seek medical attention for the experience of pain. FCT is used to address asymmetry and dysfunction through flexibility, static motor control, and dynamic motor control training. Static stretching (SS) has been shown to improve range of motion (*Konrad et al., 2023*) and has a similar effect to the foam roller included in FCT (*Mohr, Long & Goad, 2014*). However, there is currently no evidence to suggest that SS improves stability and functional patterning.

Currently, no studies investigating the effect of FCT in college students have been found. Instead, exercise interventions such as functional strength training (FST), core

training, and quadrupedal movement training (QMT) have been more extensively researched. Among these interventions, FST, consisting of lower and upper body exercises performed for 5–20 reps and 3–4 sets, twice a week over 12 weeks (*Sawczyn, 2020*), core training that includes various planks or exercises on unstable surfaces performed for 50 s or 20–30 reps for 6–8 exercises, three times a week over 6 weeks (*Šćepanović et al., 2020*), and QMT, which includes postures and movements imitating the neurodevelopmental sequence, animal postures, and movements, performed for 60 min, twice a week over 8 weeks (*Buxton et al., 2022*), have all demonstrated improvements in the FMS composite score compared to the control group. However, these interventions may not address sequentially problematic FMS sub-scores like FCT does. FCT prioritizes flexibility correction because it has a significant impact on stability and functionality. Incorrect sequencing can result in a reduction in the effectiveness of the correction. While there was a difference in physical fitness between college students and athletes, previous research has shown no difference in FMS composite scores between college athletes and general college students (*Engquist et al., 2015*). Therefore, findings from studies on college athletes can provide valuable information. For example, a study by *Bagherian et al. (2019)* found that conducting core training three times a week for 8 weeks, with each session consisting of eight exercises lasting 30 min, improved FMS composite scores of college athletes. Although there is a lack of specific studies on FCT with college students, it has been shown to be effective in improving FMS scores in various populations. These populations include high school athletes (*Song et al., 2014*), male soccer players (*Campa, Spiga & Toselli, 2019*), mixed martial arts athletes (*Bodden, Needham & Chockalingam, 2015*), firefighters (*Jafari, Zolaktaf & Ghasemi, 2020*; *Stanek et al., 2017*), and Army ROTC cadets (*Basar et al., 2019*). These studies varied in duration, ranging from 4 to 20 weeks, and frequency, ranging from two to three times per week. However, the effect of SS on the FMS has not been demonstrated. Therefore, further exploration is needed to determine if individual SS can improve FMS in college students.

Physical fitness has a strong correlation with the FMS sub-scores in college students (*Li, Tong & Chen, 2018*; *Yang, Wang & Liu, 2020*). Certain components of physical fitness were associated with symmetry and an FMS composite score of ≤14. For example, *Willigenburg & Hewett (2017)* found a strong correlation (r = 0.44) between the degree of limb asymmetry observed during the timed 6-m hop test (r = 0.44) and the magnitude of asymmetry in FMS in football players (*Willigenburg & Hewett, 2017*). A higher degree of asymmetry in females was associated with weaker abdominal muscle strength (sit-ups test; r = −0.27) and poorer flexibility (sit-and-reach test; r = 0.31) (*Koźlenia & Domaradzki, 2021*). Furthermore, research (*Koźlenia et al., 2020*) has shown significant differences in agility (agility T-test) among collegiate players categorized into groups with an FMS composite score of ≤14 and those with scores above this threshold. In addition, low-quality movement patterns in adolescent rugby union players were associated with slower sprinting speeds, poorer jumping abilities, and decreased endurance capacity (Yo-Yo intermittent recovery level 1 distance) (*Parsonage et al., 2014*). However, several studies have shown only a minimal correlation between FMS and physical fitness scores, indicating possible independence between the two (*Li et al., 2015*; *Liu, Chen & Lu, 2015*).
Although the relationship between movement patterns quality and the physical fitness of college students remains controversial, some studies have indicated a correlation between movement pattern quality and certain physical fitness components. Previous research has shown that FCT improves FMS, suggesting potential benefits for certain aspects of physical fitness. However, the effects of FCT and SS on college students' physical fitness remain unclear. Therefore, the objective of this study was to investigate the effects of 6-week FCT and SS on the movement patterns and physical fitness of college students, aiming to provide practical insights for improving movement pattern quality and overall physical fitness among this population. We hypothesized that FCT can improve the FMS composite score, asymmetry, flexibility, and sprint-related physical fitness in college students.

## MATERIALS AND METHODS

### Subjects

Forty-three male college students from a university in Guangzhou willingly took part in this research study following its promotion in university classes as part of our recruitment endeavors. The criteria used to select subjects for this study were as follows: (1) individuals between the ages of 18 and 23; (2) voluntary participation in the study; (3) no sports injuries within the past month or pain experienced during the FMS test; and (4) a score <1 or evidence of asymmetry in the FMS test. Prior to conducting the experiments, a total of forty-three male college students were subjected to the FMS test and were required to fill out a questionnaire pertaining to their basic information and history of sports injuries. This was done in order to determine if they met the specified criteria (4). Based on the predetermined inclusion criteria, a total of thirty male college students were chosen as the participants for the experimental study. Thirteen participants were excluded from the study due to specific criteria. Eight participants reported sports injuries within the past month or experienced pain during the FMS test. Additionally, five participants had a score of 1 or exhibited asymmetry.

The participants were assigned to either the functional corrective training group (FCT) or the static stretching group (SS) through the utilization of the random number table method. Two participants from the SS group withdrew from the study due to personal circumstances, resulting in their exclusion from the final analysis. Prior to the completion of the informed consent form, all participants were provided with detailed information regarding the objectives and potential risks associated with this study. Furthermore, it is important to note that this study has obtained approval from the Ethics Committee of Guangzhou Sport University, with the assigned approval number being 2021LcLL-17. This approval ensures that the study adheres to ethical guidelines and safeguards the rights and well-being of the participants involved. Basic information regarding the study subjects is provided in Table 1.

### Procedures

The tests conducted in this study encompassed the FMS and assessments of physical fitness. The aforementioned tests were performed both before and after the intervention at

**Table 1 Basic information of the study subjects.**

|  | FCT ($n$ = 15) | SS ($n$ = 13) |
|---|---|---|
| Height (m) | 1.75 ± 0.05 | 1.74 ± 0.04 |
| Body weight (kg) | 67.96 ± 9.47 | 66.90 ± 6.88 |
| BMI (kg/m$^2$) | 22.07 ± 2.33 | 21.98 ± 1.80 |
| Age (years) | 20.93 ± 0.85 | 20.85 ± 0.86 |

Note:
BMI, body mass index; FCT, functional corrective training; SS, static stretching.

the physical training laboratory of Guangzhou Sport University. To ensure the reliability of the tests, a consistent tester was assigned to administer each test. Previous studies have indicated strong intra-rater reliability (ICC-Intraclass Correlation Coefficient = 0.81–0.91) and inter-rater reliability (ICC = 0.87–0.89) among the raters (*Smith et al., 2013*). The FMS was administered by strength and conditioning instructors who possessed Level 1 and Level 2 certifications in FMS. On the first day, the participants completed the FMS and physical fitness assessments. On the second day, the participants underwent a series of physical assessments including body composition analysis, sit-and-reach test, standing long jump, 30 m sprint, and pull-ups. There was a prescribed minimum interval of 48 h between two consecutive testing sessions. The participants were given instructions to wear sneakers for both tests and were specifically advised against warming up prior to the FMS test.

## FMS test

The FMS was conducted utilizing the FMS Test Kit (Functional Movement Systems, Inc., Chatham, VA, USA). The evaluation aimed to assess the functional movement quality of the participants through the analysis of seven specific movement patterns and three pain-exclusion movements. The study included seven screening movements, namely deep squat, hurdle step, inline lunge, shoulder mobility, active straight leg raise, trunk stability push-up, and rotary stability. With the exception of the deep squat and trunk stability push-up, all movements required bilateral testing. Each movement was evaluated using a four-point scale (0, 1, 2, 3) based on established movement criteria (*Cook et al., 2014a, 2014b*). If a participant reported experiencing pain during any of the screening or pain-exclusion movements, a score of 0 would be assigned for that specific movement, and they would be advised to seek medical attention.

## Physical fitness test

*Body height:* Participants assumed a neutral stance against a wall, and their body height was assessed using a tape measure. The recorded results exhibited a precision of 0.01 m.

*Body composition:* Body composition was assessed using a body composition analyzer (InBody370; InBody, Seoul, Korea). This assessment provided measurements of body mass index (BMI) and body fat percentage (%) for subsequent analysis. Participants positioned themselves on the analyzer to initially assess their body weight. Participants proceeded by grasping held the handlebar of the analyzer for a duration of approximately 1 min in order

to obtain a measurement of body fat percentage (%). Each participant was administered a single test.

*Sit-and-reach.* Sit-and-reach flexibility was assessed by employing a sit-and-reach tester (Hebei Zhenglu Teaching Equipment Manufacturing Co., Ltd., Cangzhou city, China). Subjects assumed a seated position on the ground, extending with their legs straight and stretching them out. They then bent at the waist and proceeded to advance the cursor of the sit-and-reach tester until reaching the point of maximum bending. Results were recorded with a precision of 0.1 cm. Each participant was provided with two opportunities, and the best outcome was utilized for data analysis.

*Standing long jump.* Each participant performed a standing long jump in order to assess the horizontal distance between the starting line and the point of contact with the heel upon landing. Examiners employed a ruler to measure the distance, meticulously documenting the outcomes with a precision of 0.01 m. Participants were provided with two opportunities to execute the jump, with a 2-min interval separating each endeavor. The most optimal outcomes were chosen for analysis.

*30 m sprint:* The 30-m sprint was measured using a wireless photoelectric velocimeter (Timing Systems, Brower, USA), and the timing accuracy was recorded at 0.01 s. Subjects positioned themselves at a distance of 20 cm line, in close proximity to the timing gate, and commenced the sprint upon readiness. Participants given instructions to reduce their speed only after they had crossed the finish line, which was marked by the timing gate. Each participant underwent two trials, with a 2-min interval between each test, and the optimal outcome was documented.

*Pull-ups.* The pull-up exercise was executed using a singular bar, where participants were instructed to elevate their chin above the bar in order for the repetition to be considered valid. Subjects were instructed to perform pull-ups until reaching a point of exhaustion, with each subject being given only one test opportunity to complete the test.

## Training program

In the study, both the FCT and SS groups participated in a 6-week training program that included sessions lasting 10 to 15 min, conducted twice a week. During other periods, both groups continued their usual activities.

In line with Cook's Functional Movement System (FMS[TM]), individuals exhibiting an asymmetrical score and a score of 1 required corrective training. The correction of movement was carried out in a specific algorithmic sequence, commencing with the active straight leg raise, shoulder mobility, rotary stability, trunk stability push-up, hurdle step, inline lunge, and deep squat. Each successive correction of movement was carried out upon the conclusion of the preceding one. The progression of functional correction training followed a logical sequence, commencing with flexibility training (breathing, foam roller, and mobility), proceeding to static motor control training (motor control), and culminating in dynamic motor control training (functional patterning).

The FCT group adhered to the FMS[TM] and underwent corrective training tailored to their individual movement patterns. The training progression included flexibility training, static motor control training, and dynamic motor control training. The FCT group was
**Table 2 Training program of the FCT group.**

| Functional movements | Number of FCT (week 1–3; n = 15) | Number of FCT (week 4–6; (n = 15) | Breathing Corrective | Set × reps | Foam roller Position | Sets × durations | Mobility Corrective | Sets × reps | Motor control Corrective | Sets × reps | Functional patterning Corrective | Sets × reps |
|---|---|---|---|---|---|---|---|---|---|---|---|---|
| Active Straight Leg Raise | 4 | 2 | Crocodile Breathing Or 90/90 Breathing | 1 × 10 | Gluteal, Hip Flexors, Hamstrings | 1 × 30 s | Active leg Lowering/ Lowering to Ground | 2 × 5 | Chop from Half | 1 × 10 | Deadlift Single Leg Single | 2 × 10 |
| Shoulder Mobility | 4 | 3 | | | Deltoid, Pectoralis | 1 × 30 s | T-Spine Rotation with Rib Grab | 1 × 10 | Trunk Stability Rotation with Knees Flexed | 1 × 10 | TGU Press to Elbow | 1 × 10 |
| Rotary Stability | 7 | 7 | | | Gluteal, Hip flexors, Deltoid | 1 × 30 s | Spine Rotation with Rib Grab | 1 × 10 | Chop from Half Kneeling | 1 × 10 | Easy Rolls | 1 × 10 |
| Trunk Stability Push Up | 0 | 2 | | | Gluteal, Hip Flexors, Deltoid | 1 × 30 s | Hip Flexor Stretch | 2 × 5 | RNT Push-up | 1 × 10 | Push-up | 1 × 10 |

**Note:**
FCT, functional corrective training.

divided into different groups based on their specific movement patterns requiring correction. During the initial 3 weeks, emphasis was placed on addressing the active straight leg raise, shoulder mobility, and rotary stability movements as a priority. A reassessment of movement patterns was conducted prior to the fourth week. If asymmetry or dysfunction (sub-score < 1) persisted, the original plan was maintained. During the fourth week, trunk stability push-up correction was introduced for two subjects, while the remaining groups proceeded with their training. During the period from the fourth to the sixth week, four corrective groups were established.

Flexibility training, static motor control training, and dynamic motor control training were administered in the physical training laboratory prior to the commencement of the first and fourth weeks. If flexibility training resulted in an enhancement of the quality of functional movement (one of the sub-scores of the FMS), it was incorporated into their corrective training program. The same principle is applicable to the incorporation of both static and dynamic motor control training, provided that they enhance functional movement quality. The particular training program was developed in accordance with the guidelines outlined on the FMS website (www.functionalmovement.com), as shown in Table 2.

The SS group engaged in static stretching exercises that focused on the thoracic, gluteal, hamstring, quadriceps femoris, and gastrocnemius muscles. Each muscle was stretched for 30 s on each side, with two sets for each muscle. Both groups concurrently adhered to their individual plans for a duration of 6 weeks, engaging in the exercises during the evening.

## Statistical analysis

Statistical analyses were conducted using IBM SPSS (SPSS 25.0; IBM, New York, USA). The data were presented as the mean ± standard deviation. The Shapiro-Wilk test was used to assess the normality of the data, while Levene's test was employed to examine the

homogeneity of variances. For non-normally distributed data, the two-sample Mann-Whitney U test was used. For normally distributed data with homogeneity of variances ($p > 0.05$), a repeated measures analysis of variance (ANOVA) with a 2 (group) × 2 (time) design was performed. Bonferroni *post hoc* tests were used when an interaction effect was present. The Greenhouse-Geisser correction was applied when the assumption of sphericity was violated. Cohen's d was calculated to determine the effect size of intervention, categorized as trivial if < 0.2, small if 0.2–0.5, moderate if 0.5–0.8, and large if ≥ 0.8 (*Cohen, 1988*).

# RESULTS

## FMS score

The results of the Shapiro-Wilk test and histogram analysis indicated that the data for both the FCT and SS groups did not follow to a normal distribution. Therefore, the two-sample Mann-Whitney U test was employed. The results indicated that there were no statistically significant differences in the scores of the deep squat, hurdle step, inline lunge, shoulder mobility, active straight leg raise, trunk stability push-up, rotary stability, and composite score between the two groups in both the pre- and post-tests. In the FCT group, there were significant improvements in the hurdle step (from 2.20 ± 0.41 to 2.60 ± 0.51; Z = −2.449, $p = 0.014$), inline lunge (from 2.73 ± 0.46 to 3.00 ± 0.00; Z = −2.000, $p = 0.046$), rotary stability (from 1.20 ± 0.41 to 1.73 ± 0.46; Z = −2.309, $p = 0.021$), and composite score (from 15.73 ± 1.58 to 18.07 ± 1.79; Z = −3.316, $p = 0.001$) between the pre- and post-tests. The intergroup comparisons between the FCT and SS groups, along with the comparison of pre- and post-measurements between the two groups, are presented in Table 3.

Figure 1 illustrates a more pronounced reduction in participants' asymmetry in the FCT group in comparison to the SS group.

## Physical fitness

The physical fitness scores of the FCT and SS groups were statistically analyzed using repeated measures ANOVA.

In the post-test, significant differences were observed between the FCT and SS groups in pull-ups (F = 8.652, $p = 0.007$) and standing long jump distance (F = 7.906, $p = 0.009$). Significant improvements in pull-up performance were observed in both the FCT group (F = 6.831, $p = 0.015$, Effect size (ES) = 0.52 (moderate)) and the SS group (F = 9.454, $p = 0.005$, ES = 0.41 (small)) from the pre-test to the post-test. The FCT group showed a greater effect size compared to the SS group. There was no significant improvement in the standing long jump distance between the pre-test and post-test for either the FCT group (F = 0.378, $p = 0.544$, ES = 0.15 (trivial)) or the SS group (F = 0.060, $p = 0.809$, ES = 0.05 (trivial)). However, the effect size was marginally greater in the FCT group. Moreover, there were no significant differences between the post-test and pre-test measurements for BMI, BF, 30-m sprint, standing long jump, and sit-and-reach in both the FCT and SS groups. The physical fitness test scores for the FCT and SS groups are presented in Table 4.

**Table 3 Comparison of FMS test scores between the FCT and SS groups.**

| Functional movements | FCT (*n* = 15) | | SS (*n* = 13) | | Mann-whitney U test (between-group difference) | | | |
|---|---|---|---|---|---|---|---|---|
| | Pre-test | Post-test | Pre-test | Post-test | Pre-test | | Post-test | |
| | | | | | Z | *p* | Z | *p* |
| Deep squat | 2.13 ± 0.52 | 2.47 ± 0.52 | 2.38 ± 0.87 | 2.38 ± 0.87 | −1.55 | 0.185 | −0.157 | 0.875 |
| Hurdle step | 2.20 ± 0.41 | 2.60 ± 0.51* | 2.23 ± 0.60 | 2.23 ± 0.60 | −0.263 | 0.792 | −1.647 | 0.099 |
| Inline lunge | 2.73 ± 0.46 | 3.00 ± 0.00* | 2.77 ± 0.44 | 3.00 ± 0.00 | −0.215 | 0.830 | 0 | 1 |
| Shoulder mobility | 2.60 ± 0.51 | 2.73 ± 0.46 | 2.23 ± 0.83 | 2.38 ± 0.51 | −1.255 | 0.209 | −1.826 | 0.068 |
| Active straight leg raise | 2.33 ± 0.62 | 2.67 ± 0.62 | 2.85 ± 0.38 | 2.69 ± 0.48 | −2.397 | 0.017 | −0.117 | 0.907 |
| Trunk stability push up | 2.53 ± 0.74 | 2.87 ± 0.35 | 2.69 ± 0.48 | 2.77 ± 0.60 | −0.365 | 0.715 | −0.227 | 0.820 |
| Rotary stability | 1.20 ± 0.41 | 1.73 ± 0.46* | 1.46 ± 0.52 | 1.46 ± 0.52 | −1.451 | 0.147 | −1.442 | 0.149 |
| Composite score | 15.73 ± 1.58 | 18.07 ± 1.79* | 16.62 ± 1.89 | 16.92 ± 1.61 | −2.062 | 0.039 | −1.685 | 0.092 |

**Notes:**
* Significant difference between pre-test and post-test ($p < 0.05$).
FCT, functional corrective training; SS, static stretching.

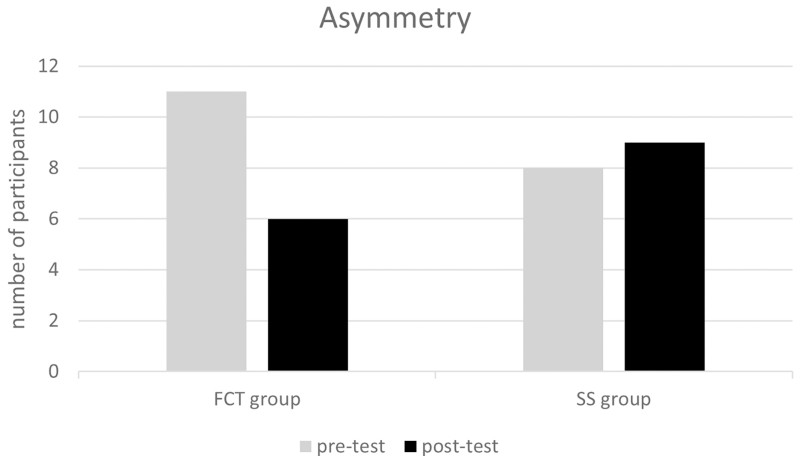

**Figure 1 The change of asymmetry in FCT and SS groups at pre- and post-test.** FCT, functional corrective training; SS, static stretching.

**Table 4 Comparison of physical fitness test scores between the FCT and SS groups.**

| Physical fitness | FCT (*n* = 15) | | | SS (*n* = 13) | | | *p*-value (between-group difference) | |
|---|---|---|---|---|---|---|---|---|
| | Pre-test | Post-test | ES | Pre-test | Post-test | ES | Pre-test | Post-test |
| BMI (kg/m$^2$) | 22.07 ± 2.33 | 22.17 ± 2.30 | 0.04 | 21.98 ± 1.80 | 21.80 ± 1.61 | −0.11 | 0.913 | 0.630 |
| BF (%) | 13.33 ± 3.11 | 12.53 ± 3.12 | −0.25 | 12.05 ± 3.76 | 11.76 ± 4.03 | −0.07 | 0.336 | 0.573 |
| 30-m sprint (s) | 4.28 ± 0.35 | 4.32 ± 0.19 | 0.12 | 4.27 ± 0.21 | 4.32 ± 0.15 | 0.28 | 0.901 | 0.96 |
| Standing long jump (m) | 2.58 ± 0.12 | 2.60 ± 0.11 | 0.15 | 2.71 ± 0.17 | 2.72 ± 0.12 | 0.05 | **0.025** | **0.009** |
| Pull-up (times) | 5.47 ± 2.80 | 6.87 ± 2.59* | 0.52 | 9.08 ± 4.19 | 10.85 ± 4.45* | 0.41 | **0.012** | **0.007** |
| Sit and Reach (times) | 14.72 ± 8.04 | 16.10 ± 7.95 | 0.17 | 17.17 ± 5.65 | 17.54 ± 6.76 | 0.06 | 0.381 | 0.622 |

**Notes:**
* Significant difference between pre-test and post-test. Differences between groups for pre-test or post-test with *p*-values less than 0.05 are indicated in bold.
BMI, body mass index; BF, body fat; FCT, functional corrective training; SS, static stretching; ES, Effect size.

## DISCUSSION

The study findings indicated that the FCT group exhibited significant improvements in hurdle step, inline lunge, rotary stability, and FMS composite score following a six-week training program conducted twice a week. Additionally, the FCT group exhibited a more pronounced reduction in asymmetries in comparison to the SS group. Nevertheless, neither group demonstrated significant enhancements in physical fitness. Our hypotheses were confirmed with regards to the FMS composite score and asymmetry, with the exception of flexibility and sprint-related physical fitness. The findings indicate that FCT has the potential to address functional movement dysfunction and asymmetries, as well as to mitigate the risk of injuries.

It is important to note that a composite score of ≤14 on FMS and the presence of asymmetry are indicative of a high risk of injuries (*Bonazza et al., 2017*; *Mokha, Sprague & Gatens, 2016*). Our findings indicated that FCT led to an improvement in the composite score and a reduction in asymmetry, thereby potentially decreasing the risk of injuries. Similar findings have been reported in previous studies involving various populations, including high school athletes, male soccer players, mixed martial arts athletes, firefighters, and Army ROTC cadets (*Basar et al., 2019*; *Bodden, Needham & Chockalingam, 2015*; *Campa, Spiga & Toselli, 2019*; *Jafari, Zolaktaf & Ghasemi, 2020*; *Song et al., 2014*; *Stanek et al., 2017*). These studies demonstrated that FCT, implemented with different frequencies and periods, resulted in enhancements in the FMS composite score and/or reduction of asymmetry. One of the studies was conducted for minimum period of 4 weeks, with four sessions per week (*Bodden, Needham & Chockalingam, 2015*). Another study had a longest period of 20-week, with a minimum of 2 sessions per week (from 12.63 ± 1.80 to 14.59 ± 0.87, +15.5%) (*Campa, Spiga & Toselli, 2019*). It appears that enhancements to the FMS composite score can only be achieved with this specific period and frequency. However, our study utilized a 6-week protocol consisting of two sessions per week, which was found to be effective in enhancing the FMS composite score (from 15.73 ± 1.58 to 18.07 ± 1.79, +14.9%). Moreover, three studies demonstrated significant improvements in stability (rotary stability and trunk stability) and advanced movements (inline lunge, hurdle step, and deep squat), while mobility (active straight leg raise and shoulder mobility) did not exhibit significant improvements. Our study yielded similar results with mean increase in stability from 3.73 to 4.60 (+23.3%) and in advanced movements from 7.06 to 8.07 (+14.3%). Among the aforementioned studies, *Basar et al. (2019)* found that the stability of cadets increased to 4.71 ± 0.62 and advanced movements improved to 7.29 ± 1.12 following a 4-week period of FCT, conducted three times per week. However, pre-test data was not provided in this study. *Campa, Spiga & Toselli (2019)* observed a 14.9% increase in stability, from 3.75 ± 0.56 to 4.31 ± 0.47, and a 28% improvement in advanced movements, from 4.59 ± 0.71 to 5.88 ± 0.66, among male soccer players following a 20-week FCT program conducted twice a week. In the study by *Stanek et al. (2017)*, firefighters exhibited enhancements in stability (4.13 ± 1.21 to 4.55 ± 0.83, 10.2%), advanced movements (4.45 ± 1.28 to 5.36 ± 1.29, 20.4%), and FMS composite score (12.09 ± 2.75 to 13.66 ± 2.28, 13.0%) following an 8-week FCT program conducted three times per week. In contrast to the

studies conducted by *Basar et al. (2019)*, *Campa, Spiga & Toselli (2019)*, and our own research, in which participants were supervised, the participants in this particular study were self-monitored. The distinction in supervision could potentially account for the comparatively lesser enhancements noted in stability and composite score.

The execution of two single-leg movements, namely the hurdle step and straight lunge, requires a certain degree of pelvic stability and core control. In this study, the FCT intervention comprised the chop from half and chop from half kneeling exercises. Both training methods are categorized as core training, and previous research has demonstrated their efficacy in improving the FMS composite score (*Bagherian et al., 2019*; *Šćepanović et al., 2020*). The improvement of the hurdle step and in-line lunge movements in the participants may be due to enhancement of the core anti-rotation ability. Regarding flexibility movements, foam roller rolling and SS are recognized as effective methods for enhancing joint flexibility. Both methods have demonstrated positive effects on joint flexibility (*Cheatham et al., 2015*). Moreover, research has indicated that there is no significant difference in the improvement of joint mobility when employing these methods individually or in combination (*Konrad et al., 2021*). The absence of a significant effect on joint mobility in this study may be attributed to the period and frequency of the intervention, which may have been insufficient for substantial improvements. Although a 6-week intervention period may be perceived as brief for improving joint flexibility, prolonging the intervention period to 8 or even 20 weeks did not result in significant improvements in joint mobility among the participants (*Campa, Spiga & Toselli, 2019*; *Stanek et al., 2017*). While *Thomas et al. (2018)* concluded in their review that a minimum of 5 min of daily stretching for at least 5 days per week is essential for achieving a positive impact on joint mobility, the studies included in the review focused specifically on lower extremity joint mobility and did not provide recommendations regarding the optimal duration of intervention cycles for improving joint mobility. In our study, the SS intervention was conducted twice a week, with each foam roller rolling and SS session lasting less than 2 min. Consequently, the limited frequency and duration of the intervention may have been the primary factors contributing to the ineffective enhancement of joint mobility. Furthermore, the presence of skeletal structure issues among the participants may have also impacted their mobility.

Although the FCT group did not show significant improvements in overall physical fitness, there was a greater degree of change in body fat percentage, 30 m sprint, standing long jump, pull-up, and sit and reach in comparison to the SS group. This result aligns with a comparable investigation conducted by *Basar et al. (2019)*, in which corrective training for FMS did not result in improvements in physical fitness outcomes. Previous studies have shown associations between the hurdle step and standing long jump, trunk stability push-up and pull-up, as well as between active straight leg raise and sit-and-reach (*Li, Tong & Chen, 2018*). Additionally, increased asymmetry has been associated with decreased flexibility (sit-and-reach test; r = 0.31) (*Koźlenia & Domaradzki, 2021*), and poor movement patterns have been linked to slower sprinting speeds and lower jumping abilities (*Parsonage et al., 2014*). Therefore, the larger effect sizes observed in the standing long jump, pull-up, and sit-and-reach in our study may be linked to the reduction in

asymmetries (from 11 to 6) and the improvement in the FMS composite score in the FCT group. The FCT group may have improved single-leg stability, balance, and hip extension by making improvements in the hurdle step and linear lunge squat. These improvements may have had an impact on the subsequent performance in the 30-m sprint and standing long jump. In addition, the incorporation of core training in the FCT group resulted in a higher energy expenditure compared to the SS group. This increased energy expenditure may potentially play a role in a slight decrease in body fat, which could account for the slight improvement in body fat percentage observed in the FCT group in comparison to the SS group.

It is important to acknowledge several limitations of this study. Despite employing random assignment, significant differences were observed among participants in the pretest for standing long jump and pull-up, potentially impacting the experimental outcomes. Furthermore, due to the diverse schedules of the participants, who were all college students, it was challenging to regulate and standardize their levels of physical activity. As a result, it is possible that participants with higher levels of physical activity demonstrated improved post-test results. Moreover, the study did not evaluate the intra-rater and inter-rater reliability of FMS testing, thereby neglecting to provide an objective demonstration of the reliability of the FMS measures. In light of these limitations, it is advisable for future studies to address potential baseline differences between groups and to systematically document and track participants' physical activity levels to facilitate a comprehensive comparison and analysis of the findings. Furthermore, it is recommended to assess the intra-rater and inter-rater reliability of FMS testing. Moreover, it is recommended to extend the application of FCT to additional populations and augment the sample size in order to broaden the scope of FCT's applicability.

The findings of this study showed that depending exclusively on FMS correction training may not be sufficient to generate significant improvements, emphasizing the necessity of simultaneous physical training. While FCT may not be considered essential, it plays a crucial role as a foundation for subsequent general and specific physical training. Previous research has extensively demonstrated the positive impact of FCT on FMS total scores among various populations. However, these studies have not specifically included college students in general. Incorporating this particular demographic in our study contributes to the current knowledge system by providing further validation of the impact of FCT on enhancing FMS total scores and reducing the injury risk. The FMS serves as a cost-effective method for evaluating injury risk and addressing asymmetries and functional deficits among college students, as it does not depend on high-tech equipment. University institutions may wish to consider acquiring an FMS kit for conducting regular screenings of students' FMS to evaluate their injury risk. Furthermore, research has shown that individuals, regardless of their prior experience with FMS testing, can effectively perform the FMS test and achieve good intra-rater and inter-rater reliability following a mere 2-h training session (*Smith et al., 2013*). By implementing corrective training based on the guidelines provided on the official website, significant improvements in FMS scores and asymmetry can be attained, resulting in a decreased risk of injuries. Combining additional physical fitness training may further optimize overall physical fitness.

## CONCLUSION

The 6-week intervention, consisting of functional movement correction training twice a week, encompassing flexibility, static motor control, and dynamic motor control training, was found to be effective in improving stability and movement patterns in male college students. Specifically, enhancements were observed in rotational stability, hurdle step, linear lunge squat, and the overall FMS score. However, the improvements in flexibility, speed, strength, and other physical attributes did not achieve statistical significance. Therefore, they should be considered as fundamental components of movement training. FMS and FCT can be effectively implemented in the college student population due to their cost-effectiveness and low learning barriers. In addition to using FMS tools for screening injury risk, FCT can also be utilized to address functional movement dysfunction and asymmetries, thereby reducing the risk of injuries.

## ACKNOWLEDGEMENTS

We would like to thank the researchers and study participants for their contributions.

### Funding

The Office of Guangdong Provincial Education Science Planning Leading Group (Theoretical and Empirical Study on the Integration of Competitive Sports Education Model into Ideological and Political Education in Higher Education Physical Education Courses 2022GXJK242) and the Guangdong Provincial Higher Education Research Project (Research on the Reform of Civics and Politics of Physical Education Curriculum in Higher Education Colleges and Universities Based on Athletic Education Model 22GZD010) funded the APC of the article. The funders had no role in study design, data collection and analysis, decision to publish, or preparation of the manuscript.

### Grant Disclosures

The following grant information was disclosed by the authors:
The Office of Guangdong Provincial Education Science Planning Leading Group: 2022GXJK242.
Guangdong Provincial Higher Education Research Project: 22GZD010.

### Competing Interests

The authors declare that they have no competing interests.

### Author Contributions

- Zhiyong Zhang conceived and designed the experiments, performed the experiments, analyzed the data, authored or reviewed drafts of the article, and approved the final draft.
- Lunxin Chen conceived and designed the experiments, performed the experiments, analyzed the data, authored or reviewed drafts of the article, and approved the final draft.

- Ziqing Qin performed the experiments, analyzed the data, prepared figures and/or tables, and approved the final draft.
- Jiaxin He performed the experiments, analyzed the data, prepared figures and/or tables, and approved the final draft.
- Chong Gao analyzed the data, authored or reviewed drafts of the article, and approved the final draft.
- Jian Sun conceived and designed the experiments, authored or reviewed drafts of the article, and approved the final draft.
- Jiancai Chen performed the experiments, authored or reviewed drafts of the article, and approved the final draft.
- Duanying Li conceived and designed the experiments, authored or reviewed drafts of the article, and approved the final draft.

### Human Ethics

The following information was supplied relating to ethical approvals (*i.e.*, approving body and any reference numbers):

GuangZhou Sport University granted Ethical approval to carry out the study within its facilities (Ethical Application Ref: 2021LcLL-17).

### Data Availability

The raw measurements are available in the Supplemental File.

### Supplemental Information

Supplemental information for this article can be found online at http://dx.doi.org/10.7717/peerj.16878#supplemental-information.

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
