# Peer review of "Effects of functional correction training on movement patterns and physical fitness in male college students"

_PeerJ, doi:10.7717/peerj.16878_

## Round 0.1 · original submission · Major Revisions

Dear Authors,

Please revise your work according to reviewers' comments or write a detailed rebuttal on a point-by-point basis.

**Language Note:** PeerJ staff have identified that the English language needs to be improved. When you prepare your next revision, please either (i) have a colleague who is proficient in English and familiar with the subject matter review your manuscript, or (ii) contact a professional editing service to review your manuscript. PeerJ can provide language editing services - you can contact us at copyediting@peerj.com for pricing (be sure to provide your manuscript number and title). – PeerJ Staff

Reviewer 1 ·

Basic reporting

Professional English is used in the article.

Context/background is not sufficiently provided (please, see my Additional comments).

Experimental design

It is not clearly stated how research fills an identified knowledge gap.

Methods are not described with sufficient detail to enable replication.

Validity of the findings

Data have been provided.

Additional comments

Introduction
- lines 38-41 – The sentence is not quite easy to understand – does it concern the failure in the Physical Education courses?
- the problem of low physical fitness of Chinese students is not explained in more detail. Also, there is no introductory mention of quality of physical functional movements in students (in China or in general) and no mention of previous findings of FMS training interventions in the student population, meaning there is no clear introduction to the problem of the study, with referral to the previous studies performed on the similar population. Please, adjust the Introduction accordingly.
- lines 64 and 66 – surnames of the cited authors should not be written with capital letters

Materials & Methods
- Subjects – participant recruitment should be explained better. It is not clear how they were initially recruited – were they invited to participate or were the volunteers recruited based on the low results on the regular screening they perform during their classes (as could be assumed based on the inclusion criterion (4) “presence of scores…”)? Please, explain in more detail.
- Procedures – it is not stated when the measurements of FMS test and physical fitness test were performed
- Training program – Table 2 is not referred to (cited) in the text. The description of the training program could be described more clearly in terms of timeline and description of the training progression.

Results
- line 152 – please, write Shapiro-Wilk instead of S-W
- line 173-174 – “In addition, there was no significant difference between the post-test and the pre-test in both the FCT and SS groups”. It is not quite clear what this sentence means, since the authors state previously in the same paragraph that “pull-ups in the FCT and SS groups showed significant improvement after corrective training”. Please, explain.

Tables
- abbreviations used in tables should be explained in the table legends, below the tables
Table 1 - please, add measurement units in the variable column (m, kg, years). The legend below the table (* Significant difference between pre-test and post-test) does not belong to this table
Table 2 – there is no title of the column 1. Also, as already mentioned, the legend below the table (* Significant difference between pre-test and post-test) does not belong to this table either
Table 4 – either significant pre-post test differences should be marked by the * in the table, or the text “* Significant difference between pre-test and post-test” is not needed in the table legend. Add explanation for the abbreviation ES (effect size) to the table legend. Modify the title of the last to columns to read “P-value (between-group difference)”

Reviewer 2 ·

Basic reporting

I like the idea of this work and appreciate the authors' efforts. However, before considering the work for publication, it needs significant improvement. The introduction is quite superficial. It lacks deeper context, explanation of the issues, and broader references to the literature. The authors should strongly emphasize the practical significance of conducting similar observations. The same goes for the discussion. The authors only briefly compare their results with the observations of other authors. The description of the materials and methods, as well as the presentation of the results, also need improvement. Below are detailed comments. I encourage the authors to consider them.

Abstract:
L17: physical functional movements – physical is not necessary. I suggest remove that phrase from the whole paper where it is repeated. Moreover, I suggest to use “movement patterns” or “movement patterns quality”.
L20 – 30 college students, and then EX=15 and control =13. Please clarify.
Also briefly describe the training modalities, and how did you measure physical fitness? Also you should provide any basic data consider your study sample: age, BMI, sex at least. Tell more, who underwent in experiment, how experiment was conducted, and what you measured.
Due to aim, when you mention “and to provide reference for improving the quality of physical functional movements and physical fitness of college students”. You should also provide this references. This issue is also repeated at the end of your paper in conclusion paragraph. So, re-write the aim or, provide missing information. I suggest fill the lack information.

Introduction:
General comment: The introduction lacks depth. You need to include a comprehensive review of the existing literature related to the research topic. Highlight the gaps in the literature that the current study aims to address. Provide a strong rationale for the study, explaining why the research is important and how it contributes to the existing body of knowledge.
L51- why you use FMS? Please clarify. Tell more about the test.
L59 -Moreover, extend the paragraph about corrective strategies to improve movement patterns. See below:
Bagherian, S., Ghasempoor, K., Rahnama, N., Wikstrom, E.A. (2019). The Effect of Core Stability Training on Functional Movement Patterns in College Athletes. Journal of Sport Rehabilitation, 28(5), 444-449.
Song, H.S., Woo, S.S., So, W.Y., Kim, K.J., Lee, J., Kim, J.Y. (2014). Effects of 16-week Functional Movement Screen training program on strength and flexibility of elite high school baseball players. Journal of Exercise Rehabilitation, 10(2), 124–130.
L64 – it is to briefly discussion about movement patterns quality and motor abilities. Please extend it. You can check below studies (Some of this study you can also use in the discussion -see further comments):
Willigenburg, N., Hewett, T.E. (2017). Performance on the Functional Movement Screen Is Related to Hop Performance But Not to Hip and Knee Strength in Collegiate Football Players. Clinical Journal of Sport Medicine, 27(2), 119-126.
Koźlenia, D., Domaradzki, J., Trojanowska, I., & Czermak, P. (2020). Association between speed and agility abilities with movement patterns quality in team sports players. Med. Dello Sport, 73, 176-186.
Silva, B., Clemente, F.M., Martins, F.M. (2018). Associations between functional movement screen scores and performance variables in surf athletes. Journal of Sports Medicine and Physical Fitness, 58(5), 583-590.
Koźlenia D, Domaradzki J. The Impact of Physical Performance on Functional Movement Screen Scores and Asymmetries in Female University Physical Education Students. Int J Environ Res Public Health. 2021;18(16):8872. Published 2021 Aug 23. doi:10.3390/ijerph18168872
Parsonage, J.R., Williams, R.S., Rainer, P., McKeown, I., Williams, M.D. (2014). Assessment of conditioning-specific movement tasks and physical fitness measures in talent identified under 16-year-old rugby union players. Journal of Strength & Conditioning Research, 28(6), 1497-1506
L66 - Before the last paragraph, reasume provided information, underline the gap in the literature and value of your study, then go the aim, and possible results (hypothesis).


Materials and Methods:
General comment: The description of the materials and methods should be detailed and clear, allowing other researchers to replicate the study.
L76 – how did you recruited your study group? How many individuals were in the initial assessment? In the finale stage how many males and females? How many individuals were excluded?
L89 – did you perform ICC for intra-rater reliability?
Koo, T.K., MY, L. (2016). A Guideline of Selecting and Reporting Intraclass Correlation Coefficients for Reliability Research. Journal of Chiropractic Medicine, 15(2), 155-163.
L78 – no injuries in whole life or some period?
L86 – in the table 1 add BMI please
L99 – add the reference for the “standards”
L101 – I miss the information about body height measures.
L141 – if you use RM-ANOVA did you check data variance and sphericity?

Results:
The results description is unclear for me. I have doubts when you refer to inter-group differences and when to intra-group differences? You provide values for ANOVA results (F and p) but where are Bonferroni test results? The appropriate changes should be introduced into table 4 when needed.
Discussion:
General comment: Compare the results with relevant studies from the literature. Discuss similarities, differences, and potential reasons for discrepancies. You could refer to previously provided studies.
L182 – but, what exactly your result mean? What observed improvements in conducted test mean? Clearly state the implications of the findings for the field of study.
L221/222- Discuss the limitations of the study and suggest directions for future research.
L227 - Relate the findings back to the practical implications mentioned in the introduction. How can the results be applied in real-world scenarios?

Conclusion:
General comment: Emphasize the practical applications and implications of the study's results.

Experimental design

All my concerns were provided in basic reporting section.

Validity of the findings

All my concerns were provided in basic reporting section.

Additional comments

References:
The references included only 22 positions. It is showing that authors should perform more extensive literature review and refer to higher amounts of studies. It will enable enhancing the manuscript scientific value.

---

## Round 0.2 · accepted · Accept

Dear Authors, your manuscript is acceptable for publication in its current form.

Reviewer 1 ·

Basic reporting

Nothing to add.

Experimental design

Nothing to add.

Validity of the findings

Nothing to add.

Additional comments

The authors addressed all the reviewers’ concerns and the article was substantially improved. The article can now be accepted.

I only point to a few sentences which still need correction due to typing errors or similar minor omissions:
Line 27 - ….were analyzed using two samples of the Mann-Whitney U test
Line 94 - These populations include including high school athletes
Line 176 - The recorded results were recorded exhibited a precision of 0.01 meters.
Line 191 - Examiners employed a ruler to employed a ruler to quantify the distance
Line 297 - mitigate reduce the risk of injuries.
Line 334 - movements in the participants may because of enhancement
Line 351 - the participants may have also impacted their mobility movement

Reviewer 2 ·

Basic reporting

All issues have been addressed. I appreciate the efforts of the authors in improving the article.

Experimental design

All issues have been addressed.

Validity of the findings

All issues have been addressed.

Additional comments

All issues have been addressed.